# Preparation and Performance Study of Photoconductive Detector Based on Bi$_2$O$_2$Se Film

Jun Liu, Zhonghui Han, Jianning Ding, Kang Guo, Xiaobin Yang, Peng Hu , Yang Jiao and Feng Teng *

School of Physics, Northwest University, Xi'an 710127, China; liujun1@stumail.nwu.edu.cn (J.L.);
202121188@stumail.nwu.edu.cn (X.Y.)
* Correspondence: tengfeng@nwu.edu.cn

**Abstract:** Bi$_2$O$_2$Se, as a novel two-dimensional semiconductor material, has been prepared and used in the field of photodetection. Herein, Bi$_2$O$_2$Se nanosheets were prepared using a hydrothermal method. Bi$_2$O$_2$Se films were also prepared using a drop-coating method. A photoconductive detector based on the Bi$_2$O$_2$Se film was constructed. The influence of nanosheet size was considered. Ultrasonic crashing treatments and different drying processes were used for the improvement of device performance. The obtained results demonstrate that the Bi$_2$O$_2$Se film based on treated nanosheets is denser and more continuous, leading to a higher photocurrent (1.4 nA). Drying in a vacuum can further increase the photocurrent of the device (3.0 nA). The photocurrent would increase with the increase in drying temperatures, while the dark current increases synchronously, leading to a decrease in the on/off ratio. The device based on Bi$_2$O$_2$Se film was dried in a vacuum at 180 °C and exhibited high responsivity (28 mA/W) and detectivity (~4 × 10$^9$ Jones) under 780 nm light illumination. Together, these results provide a data foundation and vision for the further development of photodetectors based on Bi$_2$O$_2$Se material.

**Keywords:** Bi$_2$O$_2$Se; infrared photodetector; hydrothermal; film; photoconductive



## 1. Introduction

Two-dimensional (2D) materials, due to their unique layer structures and high carrier mobility, have received much attention in the field of electronic and optoelectronic devices [1–7]. Various high-crystallinity 2D materials have been prepared and used to construct high-performance optoelectronic devices, such as MoS$_2$, InSe, MoSe$_2$, and so on [6–11]. As a new Bi-based 2D semiconductor material with good stability under air conditions, Bi$_2$O$_2$Se possesses great potential for the construction of infrared photodetectors, phototransistors, and ultrafast lasers [4,5,8,10,12]. Bi$_2$O$_2$Se has a modest bandgap of 0.8 eV, high carrier mobility (450 cm$^2$/(V·s) at room temperature) and concentration (>10$^{17}$ cm$^{-3}$), and good thermal and chemical stability [3,4,8]. Compared with other 2D layered materials, the layers of Bi$_2$O$_2$Se are connected by relatively strong electrostatic forces. In addition, due to this interlayered zipper-like structure, it exhibits many unique characteristics. However, high-quality and high-crystallinity Bi$_2$O$_2$Se nanosheets are mostly grown on mica substrate via the chemical vapor deposition (CVD) method [4,10,12,13]. The electrostatic interaction between Bi$_2$O$_2$Se and the mica substrate leads to difficulty when using transfer technology. Moreover, the CVD technology needs the support of high-vacuum and high-temperature technology [4,5,14–16]. These factors affect the development of Bi$_2$O$_2$Se-based devices and limit their actual applications. Therefore, some simple and effective methods to prepare Bi$_2$O$_2$Se-based optoelectronic devices are necessary [17–19].

The hydrothermal method is always used to prepare nanomaterials in a large scale [20–24]. Ding et al. [21] synthesized Bi$_2$O$_2$Se nanosheets using the hydrothermal method, using Bi(NO$_3$)$_3$·5H$_2$O, deionized water, hydrazine hydrate, Se powder, and LiNO$_3$:KNO$_3$ mixed salts (1:1.35 in a molar ratio) as raw materials. The obtained Bi$_2$O$_2$Se nanosheets display

superior photocatalytic performance combined with $TiO_2$ nanoparticles [21]. Yuan Sun et al. synthesized ultrathin $Bi_2O_2Se$ nanosheets using the hydrothermal method with $LiNO_3$ as an adjuvant. They carefully studied the influence of $LiNO_3$ on the morphology of the $Bi_2O_2Se$ nanosheets, and prepared large-size nanosheets (0.5–1 μm). The obtained ultrathin $Bi_2O_2Se$ nanosheets were used in photoelectrochemical tests and exhibit a high current density of 4.11 μA cm$^{-2}$ at an applied voltage of 1.0 V [25]. Leyang Dang et al. synthesized $Bi_2O_2Se$ nanosheets (size > 60.0 μm and thickness ~4.92 nm) using a hydrothermal method with an organic ion as a template and constructed a photodetector, which showed a limited responsivity of 842.91 A W$^{-1}$ and a detectivity of $8.18 \times 10^{12}$ cm Hz$^{1/2}$ W$^{-1}$ under a 532 nm laser [20]. Gexiang Chen et al. also fabricated $Bi_2O_2Se$ nanosheets using the hydrothermal method and used it to construct a photoelectrochemical-type self-powered photodetector. The responsivity and response time of the obtained device can reach 20 μA W$^{-1}$ and 0.12 s [1]. Generally, the performance of photoconductive-type photodetectors based on nanoparticles is seriously affected by the quality of the film. The different film-preparation process also affects the performance of the obtained photodetectors. As mentioned above, most of the $Bi_2O_2Se$ samples produced using the hydrothermal method are in powder form. In order to construct a photodetector, the $Bi_2O_2Se$ powder should be prepared as film [1,24,26,27]. A skillful membrane-preparation process is required for the construction of high-performance optoelectronic devices [28,29]. However, few studies have focused on the impact of film-preparation processes on the final device performance.

In this work, the $Bi_2O_2Se$ nanosheets were prepared using a simple hydrothermal method, with $Bi(NO_3)_3 \cdot 5H_2O$, $N_2H_4 \cdot H_2O$, Se powder, and NaOH used as raw materials. The morphology and composition were characterized using SEM, XPS, and XRD [16,30–35]. The obtained $Bi_2O_2Se$ nanosheets were used to construct a photoconductive-type photodetector using a drop-coating method [23,25,36]. Considering the influence of film quality on the performance of obtained devices, the size of the nanosheets and the drying process were studied in detail. The high photocurrent can be achieved by ultrasonic-crushing the nanosheets and drying them in a vacuum at high temperatures.

## 2. Materials and Methods

### 2.1. Materials

All chemicals were of analytical grade and were used as received without further purification. Bismuth nitrate pentahydrate ($Bi(NO_3)_3 \cdot 5H_2O$, 99%) was purchased from Tianjin Kemiou Chemical Reagent Co., Ltd. (Tianjin, China). Hydrazine hydrate ($N_2H_4 \cdot H_2O$, 80%) was provided by Tianjin Damao chemical reagent factory (Tianjin, China). Selenium powder (Se, 99.9%) was provided by Shanghai Macklin Biochemical Co., Ltd. (Shanghai, China). Sodium hydroxide (NaOH, 99%) was bought from Tianjin Hedong District red rock reagent factory (Tianjin, China).

### 2.2. Synthesis of Bismuth Oxyselenide

The $Bi_2O_2Se$ nanosheets were synthesized using a hydrothermal method. First, 2 mmol of $Bi(NO_3)_3 \cdot 5H_2O$ was dissolved in 10 mL of deionized water in beaker A and stirred for 0.5 h. Then, 220 μL of $N_2H_4 \cdot H_2O$, 600 mg of NaOH, and 10 mL of deionized water were mixed in beaker B and stirred to form a saturated alkaline solution. Then, 1 mmol of Se powder was put into beaker B with ultrasonic treatment until the solution became a transparent orange color. Then, the solutions in beaker A and beaker B were poured into a 50 mL Teflon vessel container and stirred for another 15 min. Then, the Teflon vessel was sealed in a stainless-steel autoclave and placed in a furnace at a temperature of 80 °C for 6 h. After the reaction was completed and the instrument naturally cooled down to room temperature, the product was washed several times with deionized water and ethanol via centrifugation, and the $Bi_2O_2Se$ nanoparticles were obtained. The schematic diagram of the preparation process is shown in Figure 1.

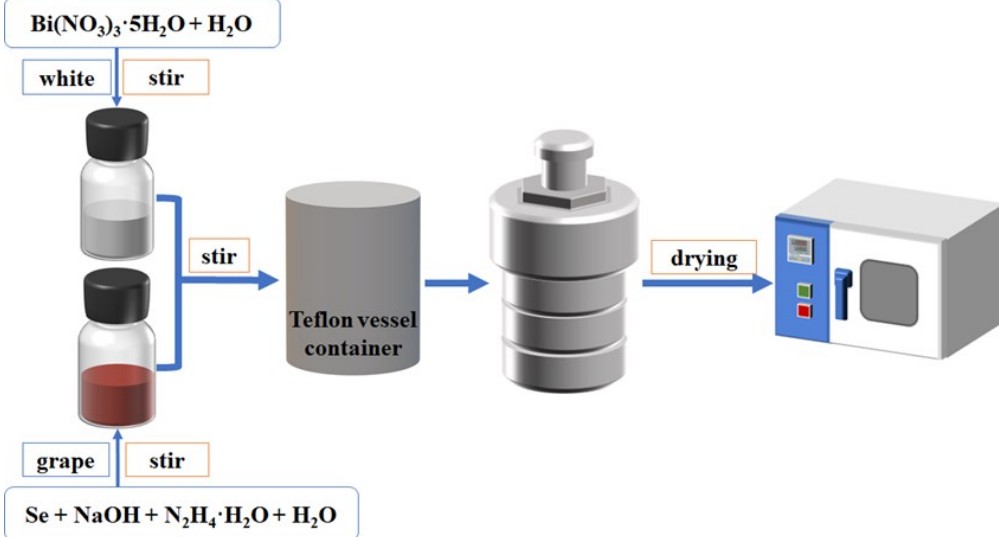

**Figure 1.** Schematic diagram of the preparation process of $Bi_2O_2Se$ nanosheets using the hydrothermal method.

*2.3. Characterization*

The crystallinity and phase were measured using an X-ray diffraction (XRD DX-2700, Hao Yuan, Dandong, China) pattern. The morphologies of the obtained samples were characterized and investigated with scanning electron microscopy (SEM, SU8020, Hitachi, Tokyo, Japan). X-ray photoelectron spectroscopy (XPS, ESCALAB 250Xi, ThermoFisher, Waltham, MA, USA) was used to analyze the chemical state of the element. UV-vis diffuse reflectance spectra of the samples were obtained using a UV-vis spectrometer (F-7000, Hitachi, Tokyo, Japan).

*2.4. Construction of Photodetectors*

First, 10 mg $Bi_2O_2Se$ nanosheets were ultrasonically dispersed in 0.5 mL deionized water for 20 min, and the $Bi_2O_2Se$ dispersion was obtained. Then, 10 μL of $Bi_2O_2Se$ dispersion was dropwise coated on the glass substrate. After drying the drip-coating solution, two silver pastes were dotted on the surface of the $Bi_2O_2Se$ film, and the $Bi_2O_2Se$ photodetector was obtained. The effective illumination area of the photodetector is $2 \times 10^{-4}$ cm$^2$.

*2.5. Photodetector Performance Test*

The steady-state electrical performance of the $Bi_2O_2Se$ hybrid photodetector was measured using a probe station with a Keysight B2901A source meter as an external power supply. A light-emitting diode (LED) lamp with a 780 nm center wavelength (7.71 mW cm$^{-2}$) was selected as the infrared light source. A DHC GCI-73 multifunctional precision electronic timer and a DHC GCI-7103 M-B shutter were used to measure the response (rise and decay) time of the photodetectors.

**3. Results and Discussion**

Figure 2a shows the XRD pattern of the $Bi_2O_2Se$ nanopowders. Two peaks, at 31.8° and 32.5°, can be observed, which are related to the preferred orientation on the (103) and (110) planes, respectively. Further, other peaks, at 23.9°, 43.7°, 46.6°, 53.1°, 56.2°, 57.9°, 68.1°, and 77.4°, correspond to the (101), (114), (200), (116), (213), (109), (208), and (310) planes of the tetragonal $Bi_2O_2Se$ (JCPDS No. 73-1316), respectively. The lattice constants are a = 3.891 Å, b = 3.891 Å, and c = 12.213 Å, and the grain size is about 9 nm, which was calculated using the Scherrer equation [18,21,22,25,37]. No obvious impurity peak could be found in the XRD pattern, implying that the obtained sample is $Bi_2O_2Se$, and that it has distinguished crystallinity. Figure 2b and c show the SEM images of the obtained

$Bi_2O_2Se$. It can be observed that the $Bi_2O_2Se$ nanoparticles are composed of many tiny nanosheets. [18,22] After statistical analysis, the average size of the nanosheets was about 180 nm, as shown in Figure 2d.

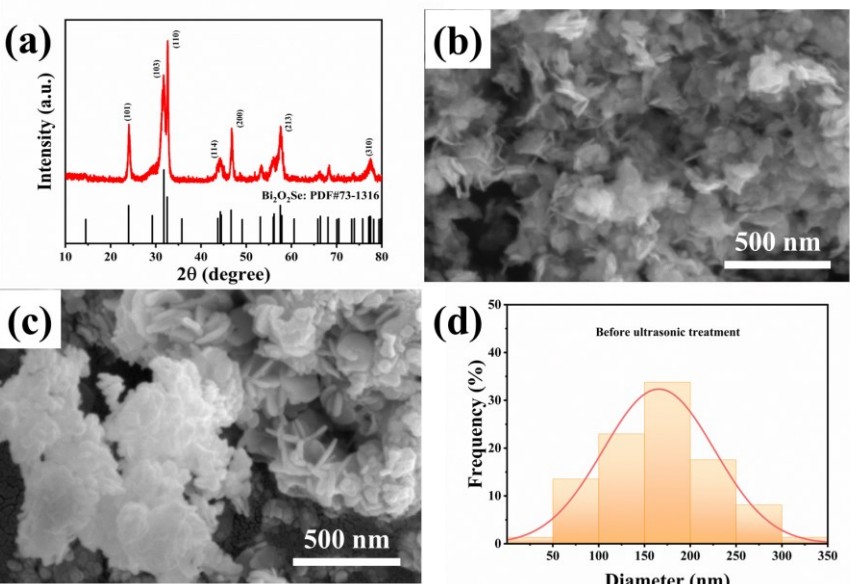

**Figure 2.** (**a**) XRD pattern and (**b**,**c**) SEM image of the obtained $Bi_2O_2Se$ nanostructures. (**d**) The columnar statistical chart of the nanosheets sizes corresponding to (**c**).

Figure 3 shows low-resolution SEM images of $Bi_2O_2Se$ nanosheets, with the corresponding EDS elemental mapping of Bi, O, and Se. As illustrated in Figure 3b–d, the distribution of elements Bi, O, and Se is uniform, and the shape is consistent with the SEM image [5,14,25]. The corresponding elemental analysis data are shown in Figure S1 and Table S1 (as seen in Supplementary Materials). It was found that the element ratio between Bi and Se is about 2:1. The oxygen content of the samples was difficult to accurately measure due to the influence of the adsorbed $CO_2$, $H_2O$, $O_2$, and organics.

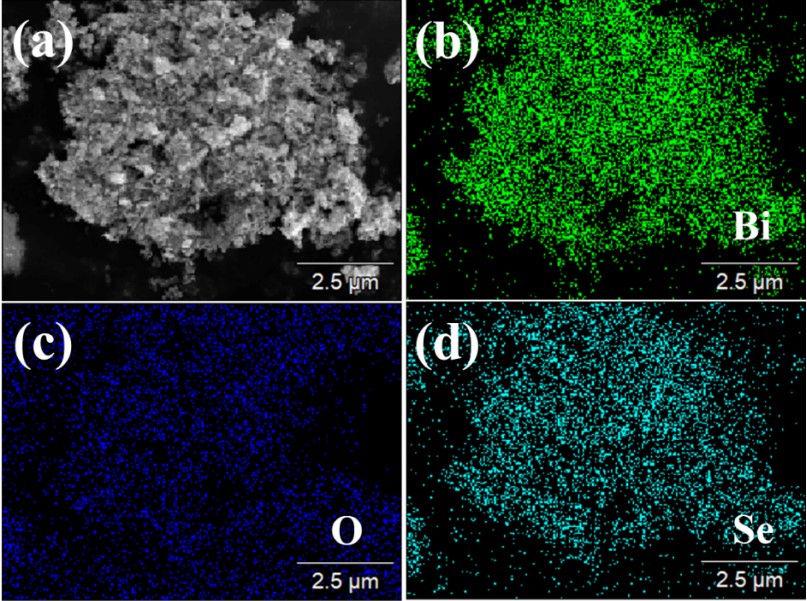

**Figure 3.** (**a**) SEM image of $Bi_2O_2Se$ nanosheets and corresponding element mapping of (**b**) Bi, (**c**) O, and (**d**) Se.

Figure 4a displays the survey XPS of the $Bi_2O_2Se$ nanosheets. The Bi, O, and Se signal can be observed clearly. Figure 3b–d show the fine spectra of elements Bi, O, and Se. In Figure 4b, Bi $4f_{7/2}$ and Bi $4f_{5/2}$ peaks, located at 158.4 and 163.7 eV, can be found, which are related to the Bi–Se bond [5]. The two higher peaks, at 159.1 and 164.5 eV, correspond to the Bi–O bond [23,25]. The O 1s spectrum (Figure 4c) comprises four peaks (529.5, 530.5, 532.2, and 534.3 eV) [20,23]. The peaks at 529.5 and 530.5 eV correlate to the Bi–O bond in the lattice and on the surface [25]. The peak at 532.2 eV corresponds to the signal of oxygen vacancies [13]. The peak at 534.3 eV may be caused by the adsorbed organics on the surface of the sample [21,38,39]. Based on the Se 3d spectra in Figure 4d, the two fitted Lorenz peaks at 52.7 and 53.6 eV are within the plausible chemical shift range of the Se 3d core level of $Bi_2O_2Se$, and correspond to Se $3d_{5/2}$ and Se $3d_{3/2}$, respectively [23,25].

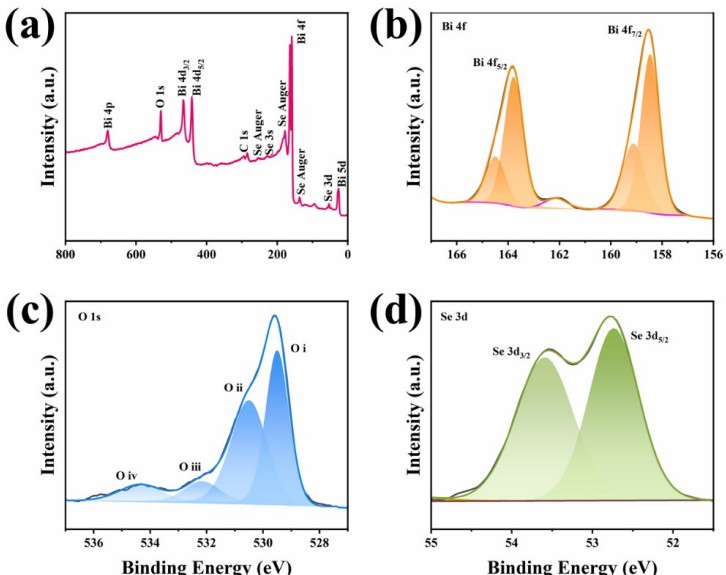

**Figure 4.** XPS of the obtained $Bi_2O_2Se$ nanosheets, (**a**) survey spectrum, and high-resolution XPS analyses of (**b**) Bi 4f, (**c**) O 1s, and (**d**) Se 3d.

The obtained $Bi_2O_2Se$ nanosheets were used to construct a photoconductive-type photodetector, which was composed of the $Bi_2O_2Se$ film and two silver pastes. The I-–V curve and I-–t curve were tested using a two-probe system on the probe station, with a Keysight B2901A source meter as an external power supply. The I-–V curve is shown in Figure 5a. The dark current of the $Bi_2O_2Se$ film photodetector is less than 0.1 nA when the bias is 15 V. Under 780 nm infrared light illumination, its photocurrent can reach 0.35 nA, demonstrating a significant infrared-light response property. In addition, from the I-–t curve shown in Figure 5b, it can be observed that the light-response characteristics are relatively stable and reproducible at 15 V bias. The stable dark current was about 0.025 nA and the photocurrent was 0.35 nA. The on/off ratio was about 10. The response speed of the obtained photodetector was also very fast.

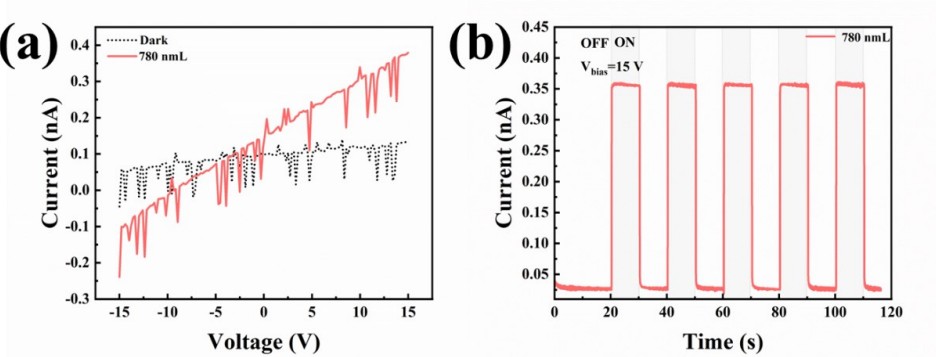

**Figure 5.** (**a**) I-—V curve and (**b**) I-—t curve of the photodetector based on $Bi_2O_2Se$ film with the applied voltage at 15 V.

Considering that the size of the nanosheets can influence the photodetection performance of the $Bi_2O_2Se$ film, ultrasonic crushing treatments were carried out to reduce the size of the nanosheets. An SEM image of the $Bi_2O_2Se$ nanosheets after ultrasonic crushing treatment is shown in Figure 6. As shown in Figure 6a–c, the size of the nanosheets reduced significantly. After statistical analysis, the average size of the nanosheets was about 60 nm, as shown in Figure 6d. Figure 7 shows the photographs and SEM images of the $Bi_2O_2Se$ film prepared using the drop-coating method with untreated and treated $Bi_2O_2Se$ nanosheets. For the film corresponding to the untreated $Bi_2O_2Se$ nanosheets (Figure 7a), cracks on the film are clearly visible, which would limit the transmission of electrons between electrodes. The rough surface and large gaps can also be observed in the SEM images, as shown in Figure 7c,e. The film prepared using the treated $Bi_2O_2Se$ nanosheets was relatively uniform and dense. No obvious cracks can be observed, as shown in Figure 7b. From the SEM images shown in Figure 7d,f, the smooth surface of the film can be observed.

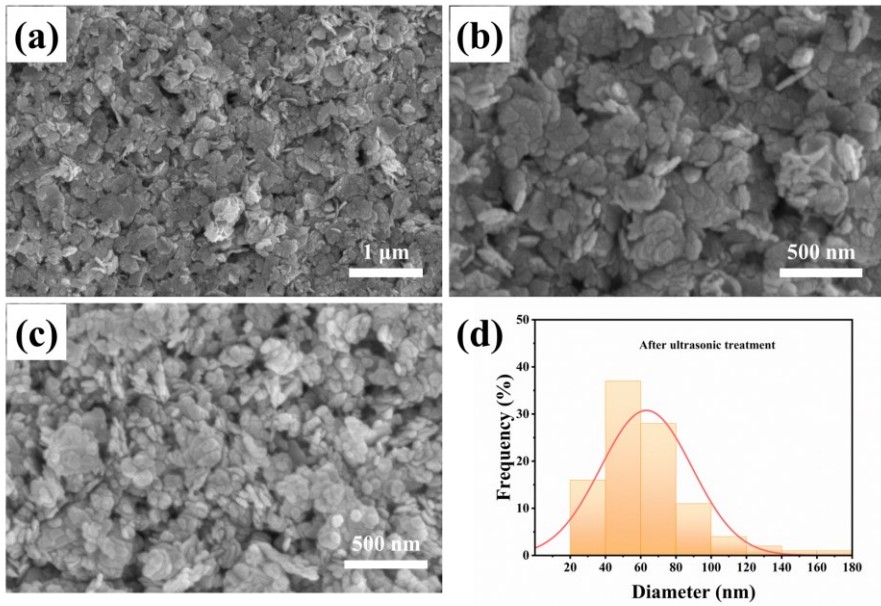

**Figure 6.** (**a**–**c**) SEM images of the $Bi_2O_2Se$ nanosheets after ultrasonic crushing treatment, and (**d**) the columnar statistical chart of the nanosheet sizes corresponding to (**c**).

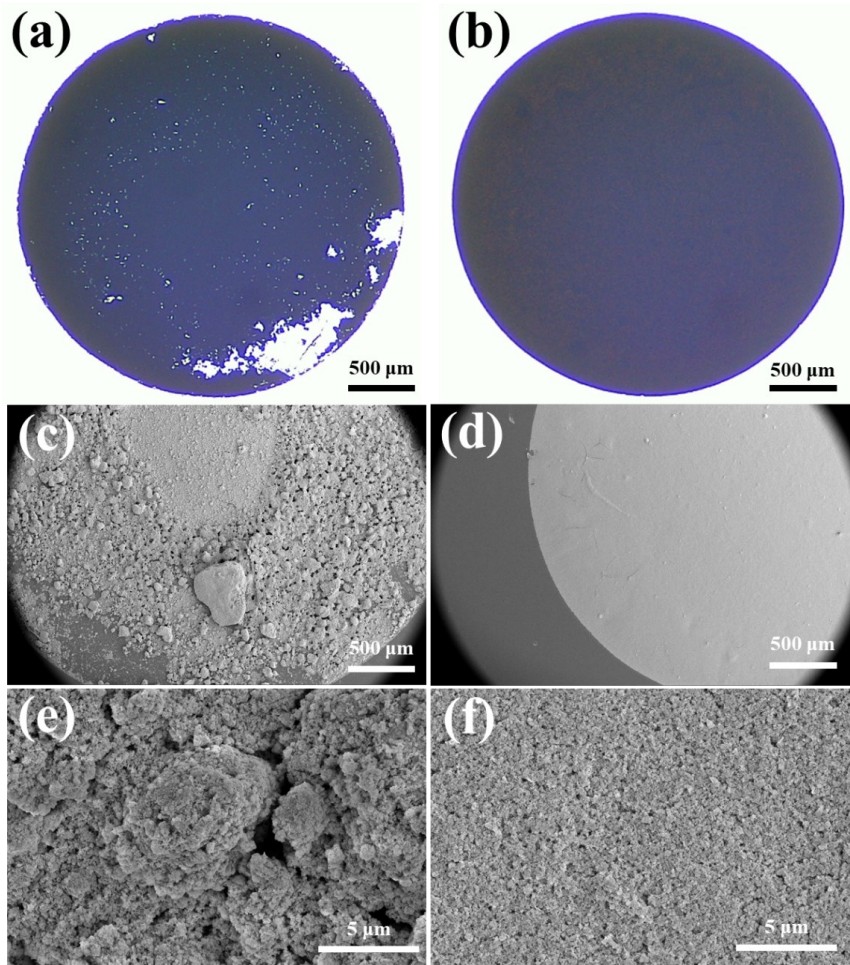

**Figure 7.** Photographs and SEM images of the films prepared using (**a**,**c**,**e**) untreated and (**b**,**d**,**f**) treated $Bi_2O_2Se$ nanosheets.

The obtained $Bi_2O_2Se$ films were also used to construct a photodetector. Figure 8 shows the performance comparison of the photodetector based on the untreated and treated $Bi_2O_2Se$ nanosheets. After reproducing the photodetector base on untreated $Bi_2O_2Se$ nanosheets, the stable dark current was 0.05 nA and the photocurrent was 0.18 nA, with an on/off ratio of about 3. The performance of the photodetector based on treated $Bi_2O_2Se$ nanosheets displayed a significant improvement. The stable dark current was 0.2 nA and the photocurrent was 1.4 nA. The increase in photocurrent was about an order of magnitude. These results demonstrate that the size of the nanosheets can indeed influence the performance of the photodetector based on $Bi_2O_2Se$ film. The improvement in performance may be caused by the change in the compactness of the $Bi_2O_2Se$ film. The smaller the size of the nanosheets, the denser the film, leading to a higher dark current and photocurrent.

Besides the size of the nanosheets, the drying condition also affects the density of the film. Subsequently, the film was dried in the air and in a vacuum at 150 °C, respectively. Figure 9 shows the comparison of the photodetectors based on $Bi_2O_2Se$ film when dried at 150 °C in the air and in a vacuum, respectively. Figure 9a,c show the I-—V curve and I-—t curve of the device dried in air, respectively. It can be observed that its photocurrent is slightly higher than the dark current at different biases. At 15 V bias, the stable dark current is about 0.02 nA, while the stable photocurrent is about 0.10 nA, with an on/off ratio of 5. Compared to the device dried in the air, the device dried in a vacuum exhibits superior photodetection performance. Figure 9b,d show the I-—V curve and I-—t curve of the photodetector dried in a vacuum. The photocurrent of this device is obviously larger than the dark current at the same bias (shown in Figure 9b). The stable dark current at

15 V bias is about 0.75 nA, and the photocurrent is about 3.0 nA. Both the dark current and photocurrent are improved, which was caused by an increase in the density of the film. The on/off ratio is about 4. Although the on/off ratio decreases, the photocurrent has increased by nearly 30 times. For comparison, the devices based on the $Bi_2O_2Se$ film that were dried at 60 °C in the air and in a vacuum were also prepared and tested. The results are shown in Figure S2 (as seen in Supplementary Materials). The I--V curves of these devices are very similar. The stable dark currents for the device dried in the air and in a vacuum are 0.07 and 0.02 nA, respectively. The corresponding photocurrents are 0.23 and 0.16 nA, respectively.

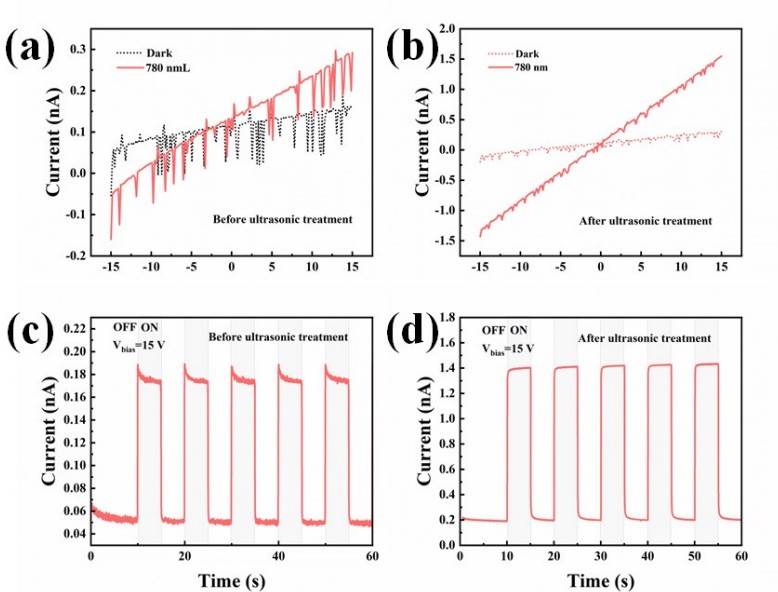

**Figure 8.** Performance comparison of the photodetector based on the (**a**,**c**) untreated and (**b**,**d**) treated $Bi_2O_2Se$ nanosheets.

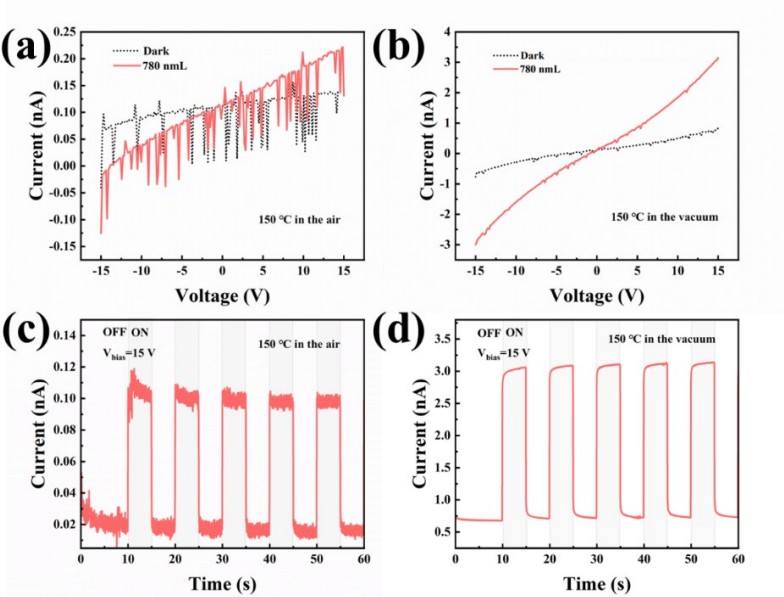

**Figure 9.** The performance comparison of the photodetectors based on $Bi_2O_2Se$ film dried at 150 °C (**a**,**c**) in the air and (**b**,**d**) in a vacuum.

The drying temperature also affects the density of the film, leading to the change in photodetection performance. The ultrasonic-crushing-treated $Bi_2O_2Se$ nanosheets were also

dried in a vacuum at different temperatures, and corresponding devices were constructed and tested. Figure 10 shows the I-−V curves and I-−t curves of the devices dried at 60, 90, 120, and 150 °C. As we expected, the dark current and photocurrent were both increased with the increase in dried temperatures, as shown in Figure 10a,b. The photocurrent of all four devices at 15 V bias is 0.2, 0.35, 1.0 and 13 nA, respectively. From the I-t curves of these devices (Figure 10c,d), it can be observed that the stable photocurrents of all four devices are 0.08, 0.23, 1.5, and 13 nA, respectively. The stable dark current is 0.01, 0.04, 0.4 and 4.5 nA, respectively. Therefore, the on/off ratios are eight, six, four, and three, respectively. As the drying temperature increases, the dark current and photocurrent increases synchronously, while the on/off ratio decreases. $Bi_2O_2Se$ film, dried at a higher temperature (180 °C) in a vacuum, was also prepared and used to construct a photodetector. Its photodetection properties are shown in Figure 11. The dark current and photocurrent are 45 and 85 nA, with an on/off ratio of less than two.

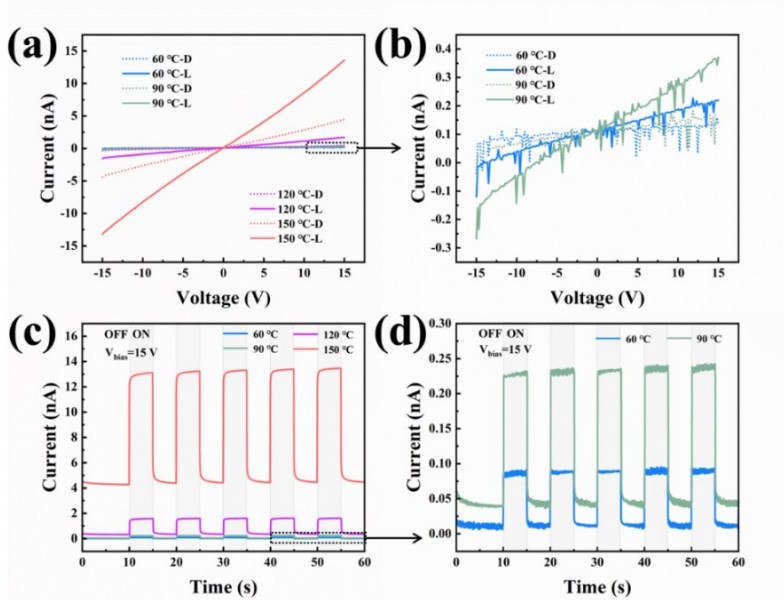

**Figure 10.** The I-−V curves (**a**,**b**) and I-−t curves (**c**,**d**) of the photodetectors based on the $Bi_2O_2Se$ film dried in a vacuum at different temperatures.

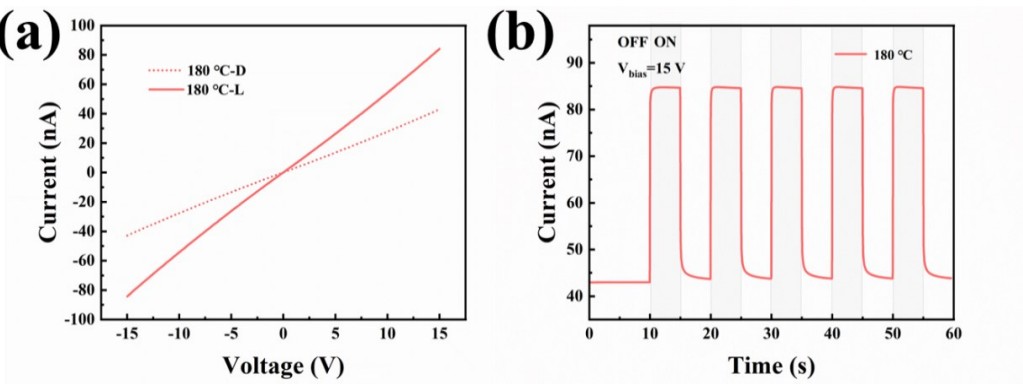

**Figure 11.** (**a**) I-−V curve and (**b**) I-−t curve of the device based on $Bi_2O_2Se$ film, dried at 180 °C in a vacuum, with the light source of 780 nm of near-infrared light.

For the device dried at 180 °C in a vacuum, more detailed tests were carried out. The obtained results are shown in Figure 12. Figure 12a shows the I-−V curves of the device under different wavelengths of illumination (from 254 nm to 1200 nm). The photocurrent under 780 nm of light illumination is the highest. Figure 12b displays the corresponding

I-−t curves of the device. The device exhibits a stable photo response under different wavelengths of illumination when the light is turned on. Figure 12c shows the I-−t curve for a single cycle under 780 nm of illumination. The rise and decay times are 100 and 500 ms, respectively, demonstrating its rapid response speed. To further evaluate the performance of the photodetector, the responsivity (R) and specific detectivity (D*) were calculated according to the following formulas [40–42]:

$$R = \left(I_{photo} - I_{dark}\right)/P \cdot S \tag{1}$$

$$D* = R(S/2eI_{dark})^{1/2} \tag{2}$$

where $I_{photo}$ is the current under light illumination, $I_{dark}$ is the current in the dark, P is the laser power density, S is the active light area, and e is the electron charge. The responsivity and detectivity of the device are shown in Figure 12d. It is obvious that the obtained $Bi_2O_2Se$ film photodetector exhibits significant near-infrared-light response characteristics. Moreover, the highest responsivity was 110 mA/W at 780 nm, with the highest detectivity being $6 \times 10^9$ Jones, which is four times higher than that at near the UV region.

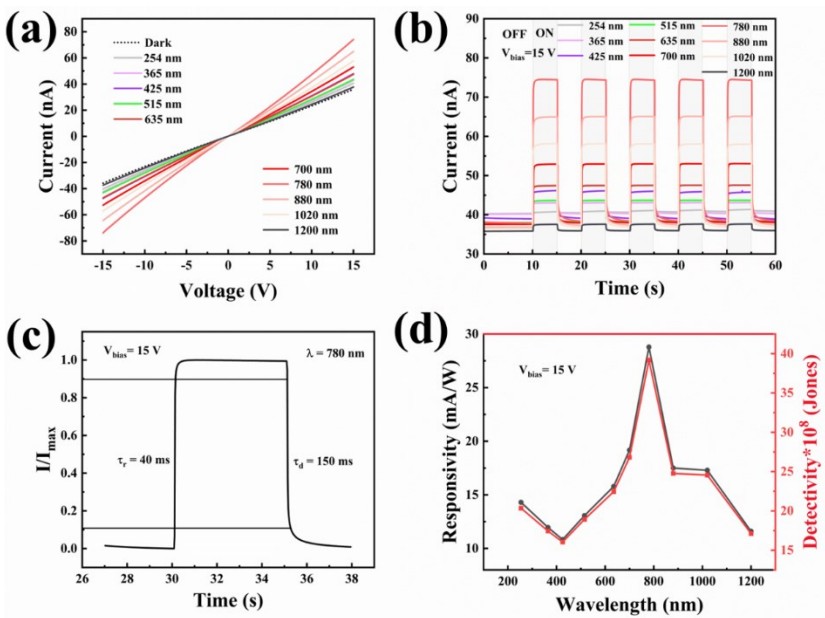

**Figure 12.** The detailed photodetection properties of the photodetector based on $Bi_2O_2Se$ film, dried at 180 °C in a vacuum. (**a**) I-−V curves, (**b**) I-−t curves, (**c**) I-−t curve for a single cycle under 780 nm of illumination, (**d**) The responsivity and the detectivity.

## 4. Conclusions

In summary, $Bi_2O_2Se$ nanosheets were prepared using a hydrothermal method with $Bi(NO_3)_3 \cdot 5H_2O$, $N_2H_4 \cdot H_2O$, NaOH, and Se powder as raw materials. The morphologies, composition, and crystallinity were characterized with SEM, XPS, and XRD. The obtained $Bi_2O_2Se$ nanosheets were prepared into a film using the drop-coating method. The $Bi_2O_2Se$ film was used to construct a photoconductive detector. Considering that the quality of the film plays an important role in the performance of the photodetector, the size of the nanosheets and the parameters of the film-preparation process were adjusted to study their influence on the performance of the devices. After the ultrasonic crashing treatment, the size of the nanosheets decreased, leading to increases in the density and continuity of the film. Furthermore, the photodetector based on the treated $Bi_2O_2Se$ nanosheets exhibits superior photodetection performance than that of untreated $Bi_2O_2Se$ nanosheets, with the photocurrent increasing from 0.18 nA to 1.4 nA. After adopting the vacuum-drying method,

the photocurrent of the device can further increase to 3.0 nA. Meanwhile, the temperature of the drying process also affects the performance of the devices. The photocurrent and dark current of the devices increased synchronously with the drying temperatures, leading to a decrease in the on/off ratio. For the device based on the $Bi_2O_2Se$ film dried in a vacuum at 180 °C, the responsivity and detectivity can reach 28 mA/W and $4 \times 10^9$ Jones under 780 nm light illumination. The obtained results can provide a foundation for the fabrication of photodetectors based on $Bi_2O_2Se$ film on a large scale.

**Supplementary Materials:** The following supporting information can be downloaded at: https://www.mdpi.com/article/10.3390/photonics10111187/s1, Figure S1: The EDX spectra show the stoichiometric ratio of $Bi_2O_2Se$ sample; Table S1: The quantitative results for $Bi_2O_2Se$ sample; Figure S2: The performance comparison of the photodetectors based on the $Bi_2O_2Se$ film dried at 60 °C, (a,c) in the air and (b,d) in a vacuum.

**Author Contributions:** Methodology, J.L. and F.T.; software, Z.H., J.D. and K.G.; validation, Z.H. and K.G.; formal analysis, J.L. and F.T.; investigation, J.D.; resources, F.T.; data curation, J.L.; writing— original draft preparation, J.L.; writing—review and editing, X.Y., P.H., Y.J. and F.T; supervision, F.T.; funding acquisition, P.H., Y.J. and F.T. All authors have read and agreed to the published version of the manuscript.

**Funding:** This work was supported by the National Natural Science Foundation of China (Nos. 51902255, 51803168, and 12004305), the Shaanxi Province Key Research and Development Projects (2022GY-356), and the Youth Innovation Team of Shaanxi Universities (No. 23JP174).

**Institutional Review Board Statement:** Not applicable.

**Informed Consent Statement:** Not applicable.

**Data Availability Statement:** Data can be provided when required.

**Conflicts of Interest:** The authors declare that they have no known competing financial interests or personal relationships that could have influenced the work reported in this paper.

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
