# Peer review of "Preparation and Performance Study of Photoconductive Detector Based on Bi2O2Se Film"

_photonics, doi:10.3390/photonics10111187_

Round 1

Reviewer 1 Report

In this manuscript, Bi2O2Se nanosheets were prepared by hydrothermal method and Bi2O2Se films were also prepared by drop coating method. It's a potential study. However, whether it is better for large-scale preparation and utilization than CVD or other methods has not been discussed

What is the classification standard for temperature in the experiment, and why these scales?

Line 9, What is CVD an abbreviation for? It is the first time.

There are some obvious English grammar mistakes in the article.

English grammar mistakes. Please check with native English speakers.

Reviewer 2 Report

1-     It was better to add the schematic diagram in the main manuscript

2-     The country of the TEM technique must be inserted

3-     The crystallite was recommended to be measured from XRD diffractogram

4-     The quantitative analysis from EDX was recommended to be inserted

5-     The novelty of this work must be cleared. Also, the different in the work to the previous similar work must be inserted in the introduction part

Reviewer 3 Report

Referee Report

On the paper “ Preparation and performance study of photoconductive detector based on Bi2O2Se film “ (photonics-2658402) by the authors Jun Liu, Zhonghui Han, Jianning Ding, Kang Guo, Xiaobin Yang, Peng Hu, Yang Jiao and Feng Teng submitted to the Photonics

This is interesting and useful paper. It reports the synthesis and investigation of the structure and optical properties of the Bi2O2Se nanosheets prepared by hydrothermal method and Bi2O2Se films also prepared by drop coating method. Photoconductive detector based on the Bi2O2Se film was constructed. The obtained results demonstrated that the Bi2O2Se film based on treated nanosheets were denser and more continuous, leading to the higher photocurrent (1.4 nA). Drying in vacuum can further increase the photocurrent of device (3.0 nA). And the photocurrent would increase with the increase of drying temperatures while the dark current increases synchronously, leading to the decrease of on/off ratio. The device based on Bi2O2Se film dried in vacuum at 180 °C exhibits high responsivity (28 mA/W) and detectivity (~ 4×109 Jones) under 780 nm light illumination. The obtained experimental results are reliable without any doubts. However, I have some questions and additions. I would like to note a few points to improve the paper before it can be published:

1.    Everything the motivation should be deleted from the Abstract.

2.    The authors should give in 1. Introduction examples of the formation of thin films:

(1). A.L. Kozlovskiy, M.V. Zdorovets, Synthesis, structural, strength and corrosion properties of thin films of the type CuX (X = Bi, Mg, Ni), J. Mater. Sci.: Mater. Electron. 30 (2019) 11819-11832. https://doi.org/10.1007/s10854-019-01556-x.

(2). A. Kotelnikova, T. Zubar, T. Vershinina, M. Panasiuk, O. Kanafyev, V. Fedkin, I. Kubasov, A. Turutin, S. Trukhanov, D. Tishkevich, V. Fedosyuk, A. Trukhanov, Saccharin adsorption influence on the NiFe alloy films growth mechanisms during electrodeposition, RSC Adv. 12 (2022) 35722–35729. https://doi.org/10.1039/D2RA07118E.

3.    For metals, their alloys, and composite samples the stoichiometry is particularly important. The deviation from stoichiometry and appearance of the oxygen anions can lead to a change in the charge state of the cations, which in turn will greatly change the electronic parameters. That will seriously affect the practical application of the materials obtained. What is the oxygen stoichiometry of prepared samples? It is well known that the complex transition metal compounds easily allow the oxygen excess and/or deficit:

(3). S.V. Trukhanov, A.V. Trukhanov, A.N. Vasiliev, A.M. Balagurov, H. Szymczak, Magnetic state of the structural separated anion-deficient La0.70Sr0.30MnO2.85 manganite, J. Exp. Theor. Phys. 113 (2011) 819-825. https://doi.org/10.1134/S1063776111130127.

(4). A. Kozlovskiy, K. Egizbek, M.V. Zdorovets, M. Ibragimova, A. Shumskaya, A.A. Rogachev, Z.V. Ignatovich, K. Kadyrzhanov, Evaluation of the efficiency of detection and capture of manganese in aqueous solutions of FeCeOx nanocomposites doped with Nb2O5, Sensors 20 (2020) 4851. https://doi.org/10.3390/s20174851.

This should be discussed in 2. Materials and Methods.

4.    The authors should mention in 1. Introduction and 3. Results and discussion such experimental techniques of non-destructive testing and determination of microstresses in materials as X-ray or/and neutron diffraction:

(5). D.I. Shlimas, A.L. Kozlovskiy, M.V. Zdorovets, Study of the formation effect of the cubic phase of LiTiO2 on the structural, optical, and mechanical properties of LixTixO3 ceramics with different contents of the X component, J. Mater. Sci.: Mater. Electron. 32 (2021) 7410-7422. https://doi.org/10.1007/s10854-021-05454-z.

(6). V.E. Zhivulin, E.A. Trofimov, O.V. Zaitseva, D.P. Sherstyuk, N.A. Cherkasova, S.V. Taskaev, D.A. Vinnik, Yu.A. Alekhina, N.S. Perov, K.C.B. Naidu, H.I. Elsaeedy, M.U. Khandaker, D.I. Tishkevich, T.I. Zubar, A.V. Trukhanov, S.V. Trukhanov, Preparation, phase stability and magnetization behavior of high entropy hexaferrites, iScience 26 (2023) 107077. https://doi.org/10.1016/j.isci.2023.107077.

5.    The proposed 6 papers should be inserted in References.

The paper should be sent to me for the second analysis after the major revisions.

Minor editing of English language required

Round 2

Reviewer 3 Report

Referee Report

On the paper “ Preparation and performance study of photoconductive detector based on Bi2O2Se film “ (photonics-2658402-v2) by the authors Jun Liu, Zhonghui Han, Jianning Ding, Kang Guo, Xiaobin Yang, Peng Hu, Yang Jiao and Feng Teng submitted to the Photonics

This paper has been well corrected and it can be recommended.

Minor editing of English language required